# Feeding Appropriate Nutrients during the Adult Stage to Promote the Growth and Development of *Carposina sasakii* Offspring

**DOI:** 10.3390/insects15040283

**Published:** 2024-04-16

**Authors:** Tong Fu, Yiran Li, Xinrun Ren, Qiao Liu, Ling Wu, Angie Deng, Ruihe Gao, Yuhong Zhang, Lina Men, Zhiwei Zhang

**Affiliations:** 1College of Forestry, Shanxi Agricultural University, Jinzhong 030801, China; ft2429547227@163.com (T.F.); erin1133@163.com (Y.L.); 15735752404@163.com (X.R.); lq3344355706@163.com (Q.L.); 17358681741@163.com (L.W.); gaoruihe1989@163.com (R.G.); linamen81@163.com (L.M.); 2Lucile Packard Children’s Hospital at Stanford, Palo Alto, CA 94303, USA; adeng11@alumni.jh.edu; 3Shanxi Dangerous Forest Pest Inspection and Identification Center, Jinzhong 030801, China; 4National Key Laboratory of Agricultural Microbiology, Biotechnology Research Institute, Chinese Academy of Agricultural Sciences, Beijing 100081, China

**Keywords:** *Carposina sasakii* Matsumura, age-stage, two-sex life table, nutritional, survival rate, fecundity

## Abstract

**Simple Summary:**

The peach fruit moth, *Carposina sasakii* Matsumura, is a significant fruit-boring pest that negatively impacts the East Asian agricultural economy. To aid the development of pest control strategies, peach fruit moths are raised and studied for multiple generations in laboratory. It is important when maintaining a laboratory moth colony to consider the effects of nutrition on the colony’s growth, survival, and reproduction. In this study, adult peach fruit moths (F_0_) were divided into separate groups, and each group was fed one of seven different nutrient solutions under laboratory conditions. The development and fitness of the moths’ offspring (F_1_) were then analyzed. The results showed that F_0_ adult peach fruit moths fed with 10 grams per liter sucrose had F_1_ offspring with significantly higher fitness and reproductive parameters, suggesting that this concentration of sucrose is more suitable for raising laboratory peach fruit moths. Thus, appropriate nutrition during the adult stage of the peach fruit moth’s life cycle could play an important role in the development of future offspring in laboratory studies and in turn influence the future of East Asian agriculture.

**Abstract:**

Nutrients consumed during the adult stage are a key factor affecting the growth, development, and reproduction of insect offspring and thus could play an important role in insect population research. However, there is absence of conclusive evidence regarding the direct effects of parental (F_0_) nutritional status on offspring (F_1_) fitness in insects. *Carposina sasakii* Matsumura is a serious, widespread fruit-boring pest that negatively impacts orchards and the agricultural economy across East Asia. In this study, life history data of F_1_ directly descended from F_0_ *C. sasakii* fed with seven different nutrients (water as control, 5 g·L^−1^ honey solution, 10 g·L^−1^ honey solution, 5 g·L^−1^ sucrose solution, 10 g·L^−1^ sucrose solution, 15 g·L^−1^ sucrose solution, and 20 g·L^−1^ sucrose solution) were collected under laboratory conditions. The growth and development indices, age-stage specific survival rate, age-stage specific fecundity, age-stage specific life expectancy, age-stage specific reproductive value, and population parameters of these offspring were analyzed according to the age-stage, two-sex life table theory. The results showed that the nutritional status of F_0_ differentially affects the growth, development, and reproduction of F_1_. The F_1_ offspring of F_0_ adult *C. sasakii* fed with 10 g·L^−1^ sucrose had significantly higher life table parameters than those of other treatments (intrinsic rate of increase, *r* = 0.0615 ± 0.0076; finite rate of increase, *λ* = 1.0634 ± 0.0081; net reproductive rate, *R*_0_ = 12.61 ± 3.57); thus, 10 g·L^−1^ sucrose was more suitable for raising *C. sasakii* in the laboratory than other treatments. This study not only provides clear evidence for the implications of altering F_0_ nutritional conditions on the fitness of F_1_ in insects, but also lays the foundation for the implementation of feeding technologies within the context of a well-conceived laboratory rearing strategy for *C. sasakii*.

## 1. Introduction

Nutritional consumption in the adult stage plays a vital role in the development, growth, and reproduction of insects [1,2]. Many species of Coleoptera, Hymenoptera, and Lepidoptera require supplementary nutrition during the adult stage to improve sexual maturity and reproduction [2]. Nutrition in the adult stage can also prolong longevity and improve the fecundity of insects [3,4]. However, previous studies have not provided conclusive evidence regarding the direct effects of adult parent (F_0_) nutritional status on the population parameters, such as rates of increase, of first-generation offspring (F_1_).

While nutritional requirements become more crucial to F_1_ after birth, minor variations during prenatal development can have significant impacts on the phenotype and adaptability of the individual [5,6]. The nutritional status of F_0_ may affect the fecundity of F_1_, as well as gamete quality and epigenetic inheritance during gametogenesis, causing transgenerational effects [7,8]. Previous research is composed of a large number of vertebrate studies. There is an absence of conclusive evidence on the direct effects of F_0_ nutritional status on F_1_ fitness in insects, though some studies speculate that the accumulation of metabolic resources in F_0_ insects can affect the fitness of F_1_ [9]. For example, adequate nutrition can significantly prolong the lifespan of F_0_ *Dolichogenidea tasmanica*, but may affect the sex ratio of its F_1_ offspring [10].

The majority of Lepidoptera species rely on carbohydrate-rich food, such as plant nectar, dew, sweat, fruit juice, or animal excrement [11], as external nutrition. Various concentrations of sucrose and honey solution are often used to study the effects of nutrition on the development and reproduction of lepidopteran F_0_ individuals [12,13]. *Plutella xylostella* adults fed with carbohydrates exhibited significantly greater longevity than those fed with water, and honey markedly increased fecundity [14]. On the other hand, *Diglyphus isaea* fed high concentrations of sugar solution showed significantly reduced adult longevity compared with those fed low concentrations of sugar solution in the same generation [15]. An absence or excess of caloric consumption seems to negatively affect insect F_0_ fitness.

*Carposina sasakii* Matsumura (Lepidoptera: Carposinidae), a species of fruit-boring moth, also named peach fruit moth, is a widespread and serious pest mainly infesting orchards in China, Japan, Korea, Southeast Asia, and the Russian Far East [16,17]. *C. sasakii* has a wide range of host plants, and its larvae can damage many kinds of fruit, including apple, jujube, hawthorn, peach, plum, apricot, and pomegranate [18,19]. The damage from *C. sasakii* larvae renders fruit inedible and in turn, negatively affects the economic income derived from fruit production. The fruit infestation rate can exceed 80% in some cases [19], even reaching 100% in poorly managed apple orchards [20]. Current research regarding *C. sasakii* focuses on life cycle, diapause, and integrated pest control. The laboratory rearing and obtaining of a substantial quantity of uniformly developed individuals is a prerequisite for population ecology, pest management, toxicity, and behavioral studies of *C. sasakii*. The caloric intake of these individuals has obvious effects on the survival of the laboratory colony. Although honey solution is often used to feed *C. sasakii* adults during laboratory rearing [21], it has been reported that adult *C. sasakii* do not feed in the field [22].

To determine which source and amount of calories are best-suited for the laboratory rearing of *C. sasakii*, varying concentrations of sucrose solution and honey solution were fed to F_0_ *C. sasakii* adults in different treatment groups in the present study. An age-stage, two-sex life table for *C. sasakii* was constructed to investigate the effects of feeding different nutrients to F_0_ on the survival rate, duration of development stages, fecundity, and life cycle of F_1_. The data collection and analysis enabled precise predictions of *C. sasakii* population growth, thereby aiding in the selection of the timing for integrated pest management strategies [23,24,25]. This study not only provides clear evidence regarding the implications of F_0_ nutritional conditions on F_1_ fitness in insects, but also lays the foundation for the implementation of feeding technologies, within the context of a well-conceived insect laboratory rearing strategy for *C. sasakii*.

## 2. Materials and Methods

### 2.1. Insects

The colony of *C. sasakii* was established from infested apples collected from a pesticide-free orchard in Taigu County, Shanxi Province, China, in the fall of 2012. Fresh, unscathed, mature Fuji apples (70–80 mm in diameter, 200–220 g in weight) were selected to raise *C. sasakii* in a laboratory incubator (MGC-250BP-2, Shanghai Yiheng Scientific Instruments, Shanghai, China) at 25.5 ± 0.5 °C, 75.0 ± 5.0% RH, with a photoperiod of 15:9 (L:D). This *C. sasakii* colony has been continuously raised in the laboratory for over 30 generations. Field individuals from pesticide-free orchards were added to the rearing colony annually to minimize inbreeding depression.

### 2.2. Methods

Newly emerged adult *C. sasakii*, with a sex ratio of 1 female:3 males [26], serving as experimental parents or the initial generation (F_0_), were placed in adult feeding containers within an incubator. Each transparent plastic adult feeding container was 7.5 cm diameter × 9.5 cm height and included an inverted vial (with a downward-facing opening plugged with a cotton ball) filled with a different nutritional solution. The bottom of the container was lined with an egg card (a sheet of filter paper roughened by a razor blade) for egg deposition. One of seven different solutions were fed to each F_0_ group, including double-distilled H_2_O (W) as the control, 5 g·L^−1^ honey solution (5H), 10 g·L^−1^ honey solution (10H), 5 g·L^−1^ sucrose solution (5S), 10 g·L^−1^ sucrose solution (10S), 15 g·L^−1^ sucrose solution (15S), and 20 g·L^−1^ sucrose solution (20S). Each treatment and control group underwent 10 replications. 

Egg cards containing first filial generation (F_1_) eggs laid within the previous 24 h were collected daily and placed into an incubated plastic container (800 mL) with a layer of cotton (5.5 g), humidified with 19.5-mL double-distilled H_2_O (ddH_2_O) at the bottom [21]. The eggs were kept in the incubation chamber for 3 d until the blackhead stage. The eggs were then separated onto smaller pieces of filter paper and placed directly on the calyx of clean apples [27], with 20 eggs per apple [21], in a transparent plastic larval feeding container (800 mL). All larval stages (first to fifth instars) were grouped into a single category, since they fed exclusively inside of the apple during the entire larval stage. ddH_2_O was sprayed daily onto the filter paper pieces containing eggs, and the paper was then covered with plastic wrap to preserve the humidity until the eggs hatched into larvae and bored into the apples. 

The emergence of mature fifth-instar larvae was checked at 19:00 every day, and these mature larvae were placed individually into transparent plastic pupation containers (11 cm diameter × 5 cm height, containing 90 g autoclaved sand and 10 g ddH_2_O), where they developed into pupae. The egg–larva period of each individual was determined based on its larval emergence date. The development of each individual was checked daily, and the dates of mature larva emergence, pupation, and adult emergence were recorded. Newly emerged F_1_ adult moths were paired by the same sex ratio as their parents of 1:3 in adult feeding containers. Because the number of male F_1_ offspring obtained from experimentally treated F_0_ adults was often insufficient for pairing, young male adults were recruited from the greater mass-rearing colony when necessary; these males, however, were excluded from life table analysis. The F_1_ generation laid eggs on egg cards that were replaced every day, and the fecundity and longevity of these adults were recorded daily at 20:00 until all F_1_ adults died. There were limitations in regards to accurately recording the exact duration time of individuals when they died in the egg, larval, and pupal stages. Thus, the duration time of eggs that did not hatch was recorded as the average egg duration time of each treatment. Similarly, the duration time of larvae that did not mature/emerge was recorded as the average larva duration of each treatment, and the duration time of pupae that did not survive to adulthood was recorded as the average duration time of pupae of each treatment.

### 2.3. Data Analysis

Raw experimental data were analyzed using TWOSEX-MSChart [28], based on the age-stage, two-sex life table theory [29,30]. A bootstrap procedure with 100,000 iterations was used to estimate the mean and standard errors of the population parameters for the offspring of each nutritional group [31]. The significant difference between each nutritional group was analyzed using the paired bootstrap test embedded in TWOSEX-MSChart software [28]. Age-stage specific survival rate (*s_xj_*), age-stage specific survival rate (*l_x_*), age-specific fecundity (*f_xj_*), age-specific fecundity (*m_x_*), age-specific maternity (*l_x_m_x_*), life expectancy (*e_xj_*), net reproductive rate (*R*_0_), intrinsic growth rate (*r*), finite rate of increase (*λ*), mean generation time (*T*), and age-stage specific reproductive values (*v_xj_*) were calculated. The parameters obtained from the life table of the age-stage, two-sex population and the TIMING-MAchart program were used to predict the population dynamics of *C. sasakii* after 90 days. Graphs were prepared with SigmaPlot 14.0 software, according to the results of the life table analysis.

### 2.4. Life Table Analysis

The calculation formulas for age-stage specific survival rate (*s_xj_*, where *x* = age and *j* = stage), age-stage specific fecundity (*f_xj_*), age-specific fecundity (*m_x_*), and age-stage specific survival rate (*l_x_*) are as follows [29]:$$s_{xj}=\frac{n_{xj}}{n_{01}}$$
$$f_{x_{j}}=\frac{E_{xj}}{n_{xj}}$$
$$m_{x}=\frac{\sum_{j=1}^{k}s_{xj}f_{xj}}{\sum_{j=1}^{k}s_{xj}}$$
$$l_{x}=\sum_{j=1}^{k}s_{xj}$$
where *n_xj_* represents the number of individuals who survive to age *x* and stage *j*; *n*_01_ represents the number of individuals at the beginning of the study as obtained from the life table; *E_xj_* is the total number of eggs laid by *n_xj_*; and *k* represents the number of stages.

The life expectancy *e_xj_* is calculated according to Chi and Su [32]:$$e_{xj}=\sum_{i=x}^{\infty }\sum_{r=j}^{k}s_{iy}^{\prime }$$
where $s_{iy}^{\prime }$ is the probability that individuals of age *x* and stage *j* will survive to age *i* and stage *y*, under the assumption that $s_{iy}^{\prime }$ = 1.

### 2.5. Population Parameters

The net reproductive rate (*R*_0_) represents the total number of offspring that an individual can produce in its lifetime:$$R_{0}=\sum_{x=0}^{\infty }l_{x}m_{x}$$

The intrinsic rate of increase (*r*) is calculated by iterative dichotomy with the Euler–Lotka formula [33]:∑x=0∞ⅇ−rx+1lxmx=1

The finite rate of increase (*λ*):$$\lambda =e^{r}$$

The mean generation time (*T*) indicates the length of time required for the population to increase by *R*_0_ times when the age distribution is stable.
$$T=\frac{lnR_{0}}{r}$$

The reproductive value (*v_xj_*) refers to the contribution of individuals in age *x* and stage *j* to the future population. According to the theory of Tuan et al. [34,35], the calculation formula is as follows:$$v_{xj}=\frac{e^{r(x+1)}}{s_{xj}}\sum_{i=x}^{\infty }e^{-r(i+1)}\sum_{y=j}^{k}s_{iy}^{\prime }f_{iy}$$

## 3. Results

### 3.1. Developmental Duration and Longevity

Differences in the effects of feeding different nutrients to adult stage F_0_ *C. sasakii* on their F_1_ offspring were observed at all stages of F**_1_** development (Figure 1, Table A1). Compared to the double-distilled water control (W) treatment, F**_0_** adults fed with honey or sucrose solution produced F**_1_** offspring with significantly shorter egg–larva durations, accelerated egg–larva growth rates (Figure 1A), and improved egg–larva survival rates (Figure 1B). The shortest F**_1_** egg–larva duration was observed in the 10 g·L^−1^ sucrose solution (10S) treatment (26.02 d), whereas the longest F**_1_** egg–larva duration was observed in the W treatment (30.36 d) (Figure 1A). Treatment groups fed with sucrose or honey exhibited a significantly prolonged F**_1_** pupa duration compared to that of the control (Figure 1C). However, feeding carbohydrate nutrients can significantly shorten F**_1_** preadult duration compared to that of the W treatment. The F**_1_** preadult duration was shorter in the 5 g·L^−1^ honey solution (5H) and 5 g·L^−1^ sucrose solution (5S) treatments (38.00 d and 38.32 d, respectively) (Figure 1E) than in the other treatments. The F_1_ preadult survival rate (*s_a_*) in the 10 g·L^−1^ honey solution (10H) treatment was higher than that of the W treatment (Figure 1F). Female F_1_ adult longevity was greater than that of males in all treatments except for the 10H and 15 g·L^−1^ sucrose solution (15S) treatments (Figure 1G,H). Moreover, F_1_ female and male adult longevities were greater in the 5S (10.55 d, 8.00 d) and 10S treatments (8.94 d, 8.17 d) (Figure 1G,H).

### 3.2. Age-Stage Specific Survival Rate

The age-stage specific survival rate (*s_xj_*) not only provides a detailed description of the survival probability of newly laid eggs at age *x* and stage *j*, but also gives a comprehensive account of stage differentiation. Differences in the age-stage specific F_1_ survival rates were observed (Figure 2). A clear overlapping phenomenon between different stages was observed due to the variation in developmental rates among F_1_ individuals. The *s_xj_* of the F_1_ pupa stage in the 10S treatment peaked (0.13) the earliest amongst treatment groups at age 31 d (Figure 2B). The peak values of *s_xj_* for the pupa stage and for F_1_ female and male adults in the 10H treatment were significantly higher compared to those of other groups (Figure 2B–D).

### 3.3. Age-Stage Specific Fecundity and Fertility

Both the proportion of female adults (*N_f_*/*N*) and the proportion of reproductive female adults (*N_fr_*/*N*) were high among the F_1_ offspring derived from non-control F_0_ adults. The 10H treatment showed the highest *N_f_*/*N* (10.67%) and *N_fr_*/*N* (9.67%) in the F_1_ among the seven treatments (Figure 3A,B). The female adult fecundity and oviposition days (*O_d_*) of F_1_ were significantly different between treatments (Table 1). Interestingly, F_1_ female adult fecundity was significantly higher in the 10S treatment (222.53) than in the other treatments, whereas F_1_ female adult fecundity was the lowest in the 10H treatment (77.63). Surprisingly, F_1_ female adult fecundity in the control treatment was higher than that of some honey and sucrose fed insects. The number of F_1_ oviposition days (*O_d_*) was significantly greater in the 5S and 10S treatments (7.64 d and 6.71 d, respectively) compared to other treatments as well. In terms of the F_1_ total pre-oviposition period (TPOP) and the adult pre-oviposition period (APOP), significant differences were observed among the seven treatments. F_1_ APOP was significantly longer in the 10H treatment (1.62 d). F_1_ APOP in the 10S treatment (1.18 d) was shorter than that in the control treatment (1.28 d), and the 10S F_1_ TPOP (39.53 d) was significantly shorter than that of the control (43.72 d).

The age-specific survival rate (*l_x_*) represents the cumulative survival probability of an individual from birth to age *x*, calculated as the sum of age-specific survival rates across all life stages (*s_xj_*). It denotes the likelihood that a newborn individual will survive to age *x* (Figure 4) [36]. The *l_x_* of F_1_ began to decline as early as age 4 d in 10S, then gradually decreased to zero by age 68 d (Figure 4A). Generally, as the insects matured, there was a clear initial increase, followed by a subsequent decrease to zero in the age-stage specific fecundity (*f_x_*_3_, adult female is the third life stage), age-specific fecundity (*m_x_*), and age-specific net maternity (*l_x_m_x_*) across the seven *C. sasakii* colonies (Figure 4). The maximum F_1_ *f_x3_* (105.00) occurred at age 30 d, and the maximum F_1_ *m_x_* (77.00 eggs) occurred at age 61 d in 10S (Figure 4B). F_1_ *m_x_* shows that *C. sasakii* reproduction began at age 30 d in the 10S treatment, while that of the W treatment began at age 39 d (Figure 4C). The maximum value of F_1_ age-specific net maternity (*l_x_m_x_*) was lower in 10S (0.88 eggs) compared to that of the control (1.29 eggs) (Figure 4D). Furthermore, fluctuations in the fecundity curve suggested that the emergence and oviposition of F_1_ individuals did not occur at a specific age.

### 3.4. Age-Stage Life Expectancy

The age-stage life expectancy (*e_xj_*) values for all F_1_ offspring gradually decreased to zero as age increased (Figure 5). Notably, the life expectancy curve of the seven treatments showed that F_0_ diets had varied effects on the growth and development of F_1_ at different stages. The initial *e_xj_* values of F_1_ for 5H and 10S treatments (18.99 days and 22.48 days, respectively) were significantly lower than those of the other treatments (Figure 5A). Furthermore, this result is consistent with the trends of mean longevity in the corresponding treatments. In addition, the *e_xj_* values of female F_1_ adults were typically higher than those of males (Figure 5C,D). The *e_xj_* curve exhibited varying degrees of fluctuation at different stages in different treatments, with higher levels of fluctuation indicating higher mortality rates.

### 3.5. Age-Stage Specific Reproduction Values

The age-stage specific reproductive value (*v_xj_*) indicates the magnitude that an individual at a particular age (*x*) and stage (*j*) contributes to the future population (Figure 6). *v_xj_* values increased significantly when F_1_ began to lay eggs. The *v_xj_* of female F_1_ adults had two peaks, observed at age 30 d (295.04) and 61 d (239.64) in the 10S treatment. The peak values of F_1_ *v_xj_* in 10S were higher than that of other treatments except 5H (319.19). A single F_1_ *v_xj_* peak was observed in all treatments except 10S. Additionally, amongst these 7 treatments, the longest reproductive period of female F_1_ was observed in 10S (32 d), which far exceeded that of W (19 d) (Figure 6C).

### 3.6. Population Dynamics Parameters

There were significant differences in the intrinsic rate of increase (*r*), finite rate of increase (*λ*), net reproductive rate (*R*_0_), and mean generation time (*T*) of F_1_ among the seven treatments (Table 2). The finite rate of increase (*λ*) of F_1_ for each treatment group exceeded 1, indicating that the F_1_ colony in all treatments experienced quantifiable growth. The highest F_1_ values of *r*, *λ*, and *R*_0_ were observed in 10S (*r* = 0.0615, *λ* = 1.0634, *R*_0_ = 12.61), indicating that F_1_ could achieve the highest colony growth with the 10S treatment. The lowest F_1_ values of *r* (0.0449) and *λ* (1.0459) were observed in 15S. The lowest F_1_ *R*_0_ (7.41) was observed in 5S (Figure 7). The F_1_ mean generation time (*T*) was significantly longer in the 10H treatments. However, the F_1_ mean generation times (*T*) in 10S (41.21 d) and 5H (40.71 d) were relatively short.

### 3.7. Simulation of Population Growth Dynamics

The population dynamics of F_1_ *C. sasakii* derived from F_0_ adults treated with seven different nutrients were predicted over a 90-day period using TIMING-MSChart (Figure 8). The colonies in all treatments showed numerical growth. Each treatment group started with 10 eggs, and the fastest-growing colony was produced from 10S F_0_ adults. At 90 days, the third-generation filial pupae (F_2_) appeared in the 10S treatment, with overlapping generations, while the F_2_ individuals in all other treatments were still in the egg–larva stage. After 90 days, the 10S treatment colony was expected to reach a total of 915.63 offspring individuals (906.02 egg–larvae, 5.67 pupae, and 3.93 adults), which was nearly twice that of predicted total of 506.55 W offspring individuals (496.27 egg–larvae, 4.44 pupae, and 5.84 adults).

## 4. Discussion

In the results of this study, the longevity, egg–larva duration, pupa duration, adult duration, and fertility of F_1_ *C. sasakii* offspring were significantly affected by both the type and concentration of nutrient solutions fed to F_0_ parents. When F_0_ adults were fed sucrose and honey solutions, the F_1_ egg–larva duration was significantly shortened, and the F_1_ egg–larva survival rate was significantly increased. A similar effect was also found in *Plodia interpunctella*, where a high-quality F_0_ larval diet accelerated F_1_ development [37]. Previous research has found that supplementation with multiple forms of carbohydrates, such as fructose, glucose, and sucrose, was sufficient in increasing *Cnaphalocrosis medinalis* F_0_ adult longevity and fecundity [12]. *Spodoptera exempta* F_0_ adult longevity and fecundity were also significantly reduced when females were fed diets lacking carbohydrates [4]. Greater longevity in adult insects may benefit mating and the oviposition period. However, these studies do not provide conclusive evidence regarding the potential effects of F_0_ adult nutrition on F_1_ adult longevity. Through this study, we noted that feeding suitable concentrations of sucrose or honey solutions can prolong the adult duration of F_1_ *C. sasakii offspring*. Our results indicate that the nutritional condition of F_0_ parents can improve F_1_ egg quality and shorten F_1_ preadult duration. Improved nutritional conditions of F_0_ can accelerate the development of F_1_ and thus benefit overall insect population growth.

It is important to note that not only does insufficient caloric consumption seem to negatively affect *C. sasakii* colonies, but excessive caloric consumption does so as well. Compared to the control group fed exclusively with distilled water, higher concentrations of carbohydrate solutions shortened the adult longevity of both female and male F_1_ *C. sasakii* and reduced F_1_ female adult fecundity. A previous study of *Diglyphus isaea* [15] observed similar phenomena, with higher concentrations of sugar solution fed to F_0_ significantly shortening adult longevity of the same generation. Therefore, we speculate that these results are related to the structure of the adult mouthparts of *C. sasakii* [22]. Nutrient solutions with higher carbohydrate concentrations, which are more adherent to surfaces and therefore more difficult to consume, may lead to an increase in internal osmotic pressure within the body of an insect and thereby affect physiological processes. Conversely, clear water and lower concentrations of carbohydrates seem to be more beneficial to *C. sasakii* growth and development. In other words, only the appropriate amount of parental caloric consumption is beneficial to producing the next generation of *C. sasakii*.

Nutrition is important for the reproductive capacity of female adults. Feeding on nutrients during the adult stage plays a major role in converting potential fecundity into actual fecundity [38]. However, there is an absence of conclusive evidence on the direct effects of F_0_ parental nutritional status on F_1_ offspring in insects. Our results show that nutrition at an appropriate concentration of honey or sucrose during the adult stage of F_0_ *C. sasakii* can increase female adult fecundity and prolong the *O_d_* of F_1_ offspring. In a previous study that fed only 10 g·L^−1^ honey solution to F_0_ adults saw improved F_1_ adult fecundity, to a certain extent [21]. A higher nutritional status of F_0_ *Chironomus tepperi* also significantly increased the fecundity of F_1_ [9]. We speculate that the adequate nutritional status of *C. sasakii* parents will enhance the reproductive potential of their offspring.

Previous research has shown that some insect species are capable of mating and laying eggs immediately after adult emergence, while other species require additional nutrients and time for reproduction [39,40,41,42]. In the present study, the APOP for newly emerged individuals across seven treatments was 0.63 to 1.62 days, indicating that *C. sasakii* adults require additional time for reproduction after emergence. This phenomenon may be associated with further post-emergence ovariole development to facilitate reproduction [43]. The APOP and TPOP of F_1_ *C. sasakii* during the adult stage were significantly shorter in suitable honey and sucrose treatments than in the control. On the contrary, it has been shown that the APOP of *Spodoptera exigua* fed with water is shorter than that with honey solution [38]. Insects may have different nutrient requirements at different life stages, and different nutrients may have varying effects on insect life activities. Further research is needed.

Both F_0_ and F_1_ adult *C. sasakii* in this study were able to successfully mate and spawn, and their eggs hatched normally when provided with only distilled water. This evidence indicates that the nutrition required for reproductive development before adult emergence was sufficient. This may also be a contributing factor to the widespread agricultural damage caused by *C. sasakii*. The results of previous studies on other moths, including *Athetis lepigone* [44], *Spodoptera exigua* [38], *Spodoptera frugiperda* [45], and *Stenoma catenifer* [46], were consistent with this phenomenon. Nutrition is considered inessential to the mating behavior of *C. sasakii*, but significantly higher adult fecundity of F_1_ female *C. sasakii* was derived from F_0_ adults that underwent nutritional treatment. Therefore, adult nutrition still influences future offspring’s fecundity and thereby, affects colony growth.

The quality of the parental insect diet has been previously demonstrated to impact the colony structure of the offspring. Ant colonies that consumed carbohydrate-rich diets showed higher numbers of worker and brood production compared to those on carbohydrate-deficient diets [47,48]. Life table analysis revealed that the *C. sasakii* colony raised on 10 g·L^−1^ sucrose solution exhibited the highest *r*, *λ*, and *R*_0_ values. Life table parameters *r*, *λ*, and *R*_0_ are used to estimate the growth and reproductive potential of insect populations [49]. Our analysis indicates that the experimental colony displayed either higher fecundity or a faster rate of development [50].

Population projection based on the age-stage, two-sex life table can reveal changes in stage structure during population growth. Understanding stage structure and colony growth predictions is crucial for pest management and laboratory colony rearing because the colony growth of *C. sasakii* varies with different external environments. In previous studies, *C. sasakii* exhibited faster colony growth with lower larval density [21]. Suboptimal nutrient concentrations lead to weakened colony growth of *C. sasakii*. The source and concentration of nutrients consumed by *C. sasakii* adults can have significant positive effects on the population growth of future generations. Understanding how F_0_ nutritional conditions affect the fitness of F_1_ is important for the development and implementation of pest control against *C. sasakii* based on feeding technologies.

## Figures and Tables

**Figure 1 insects-15-00283-f001:**
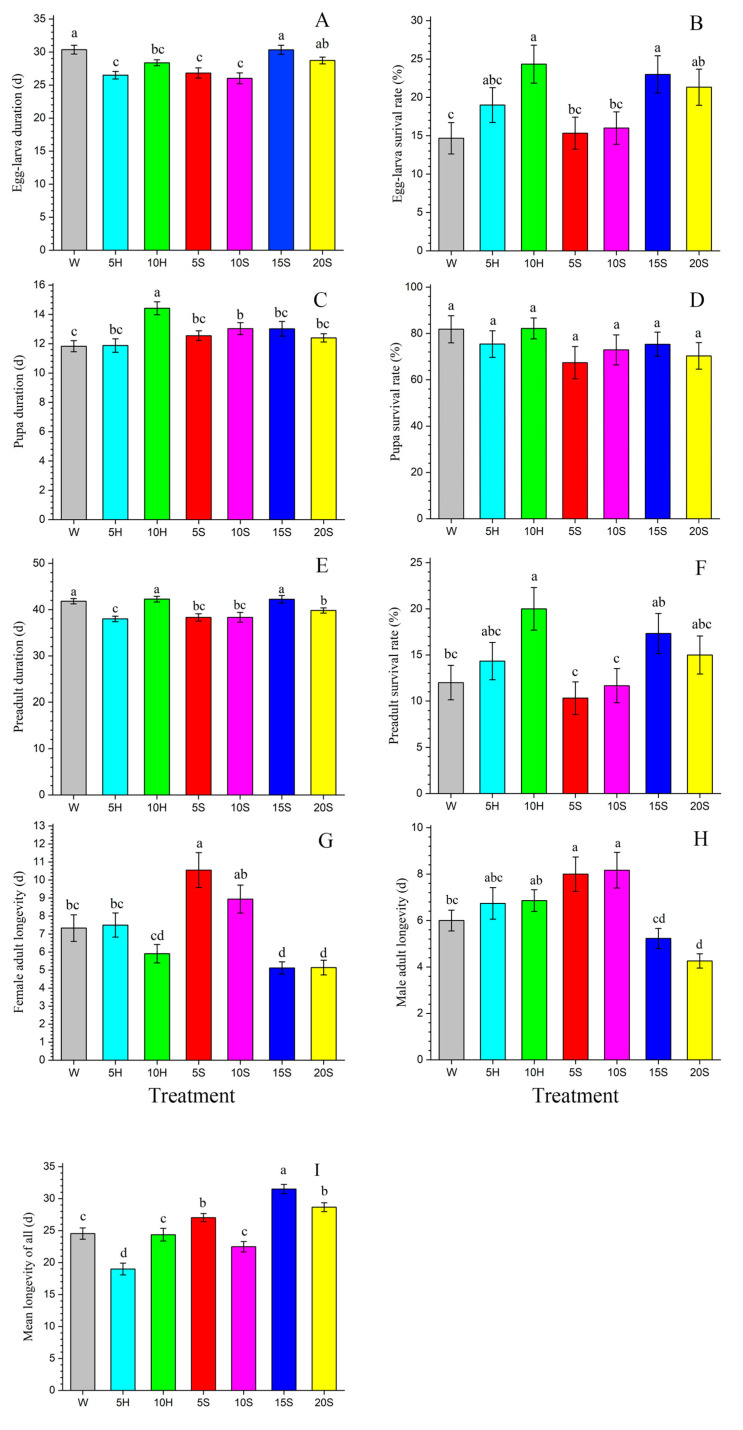
The developmental durations and survival rates of F_1_ *C. sasakii* offspring produced from F_0_ adults fed with seven different nutrient solutions. (**A**) Egg–larva duration, (**B**) egg–larva survival rate, (**C**) pupa duration, (**D**) pupa survival rate, (**E**) preadult duration, (**F**) preadult survival rate, (**G**) female adult longevity, (**H**) male adult longevity, (**I**) mean longevity of all. W.—double-distilled water; 5H.—5 g·L^−1^ honey solution; 10H.—10 g·L^−1^ honey solution; 5S.—5 g·L^−1^ sucrose solution; 10S.—10 g·L^−1^ sucrose solution; 15S.—15 g·L^−1^ sucrose solution; 20S.—20 g·L^−1^ sucrose solution. Different lowercase letters indicate significant differences between treatments (paired bootstrap test, *p* < 0.05). Standard errors were estimated by using 100,000 bootstraps.

**Figure 2 insects-15-00283-f002:**
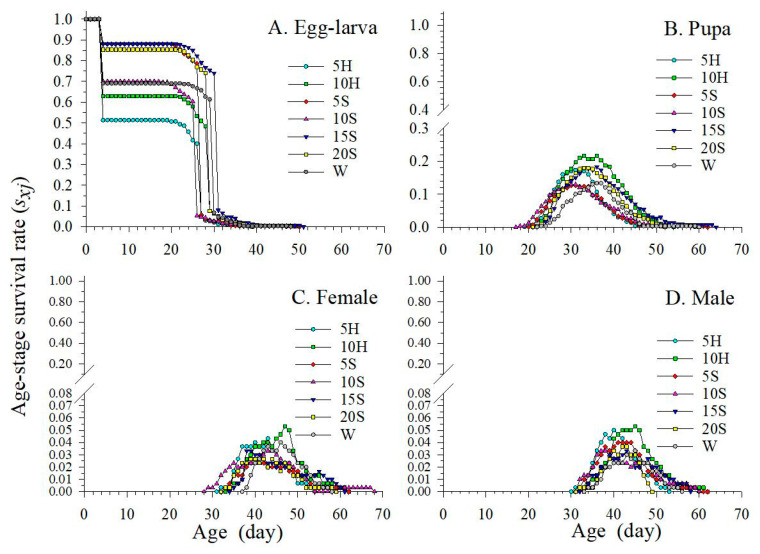
Age-stage survival rate of F_1_ *C. sasakii* produced from F_0_ adults fed with seven different nutrient solutions. (**A**) Age-stage survival rate of egg–larva, (**B**) age-stage survival rate of pupa, (**C**) age-stage survival rate of female adults, (**D**) age-stage survival rate of male adults. W.—double-distilled water; 5H.—5 g·L^−1^ honey solution; 10H.—10 g·L^−1^ honey solution; 5S.—5 g·L^−1^ sucrose solution; 10S.—10 g·L^−1^ sucrose solution; 15S.—15 g·L^−1^ sucrose solution; 20S.—20 g·L^−1^ sucrose solution.

**Figure 3 insects-15-00283-f003:**
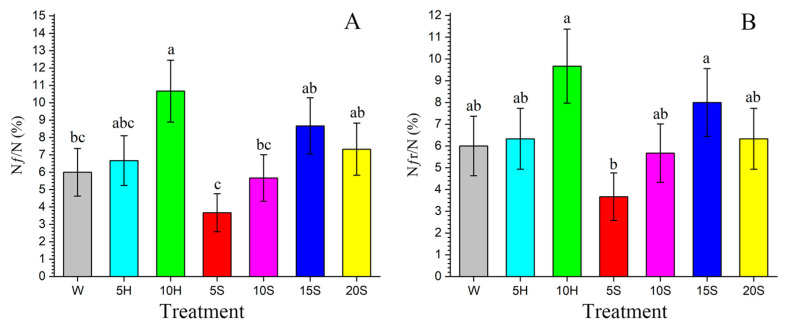
The proportion of female adults (*N_f_*/*N*) and the proportion of reproductive female adults (*N_fr_*/*N*) of F_1_ *C. sasakii* offspring produced from F_0_ adults fed with seven different nutrient solutions. (**A**) The proportion of female adults, (**B**) the proportion of reproductive female adults. W.—double-distilled water; 5H.—5 g·L^−1^ honey solution; 10H.—10 g·L^−1^ honey solution; 5S.—5 g·L^−1^ sucrose solution; 10S.—10 g·L^−1^ sucrose solution; 15S.—15 g·L^−1^ sucrose solution; 20S.—20 g·L^−1^ sucrose solution. Different lowercase letters indicate significant differences between treatments (paired bootstrap test, *p* < 0.05). Standard errors were estimated by using 100,000 bootstraps.

**Figure 4 insects-15-00283-f004:**
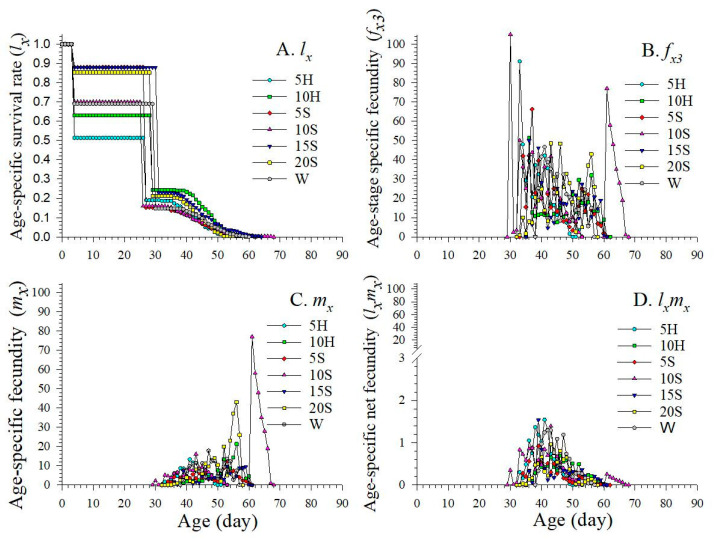
Age-specific survival rate (*l_x_*), age-stage specific fecundity (*f_x3_*), age-specific fecundity (*m_x_*), and age-specific net fecundity (*l_x_m*_x_) of F_1_ *C. sasakii* produced from F_0_ adults fed with seven different nutrient solutions. (**A**) Age-specific survival rate, (**B**) age-stage specific fecundity, (**C**) age-specific fecundity, (**D**) age-specific net fecundity W.—double-distilled water; 5H.—5 g·L^−1^ honey solution; 10H.—10 g·L^−1^ honey solution; 5S.—5 g·L^−1^ sucrose solution; 10S.—10 g·L^−1^ sucrose solution; 15S.—15 g·L^−1^ sucrose solution; 20S.—20 g·L^−1^ sucrose solution.

**Figure 5 insects-15-00283-f005:**
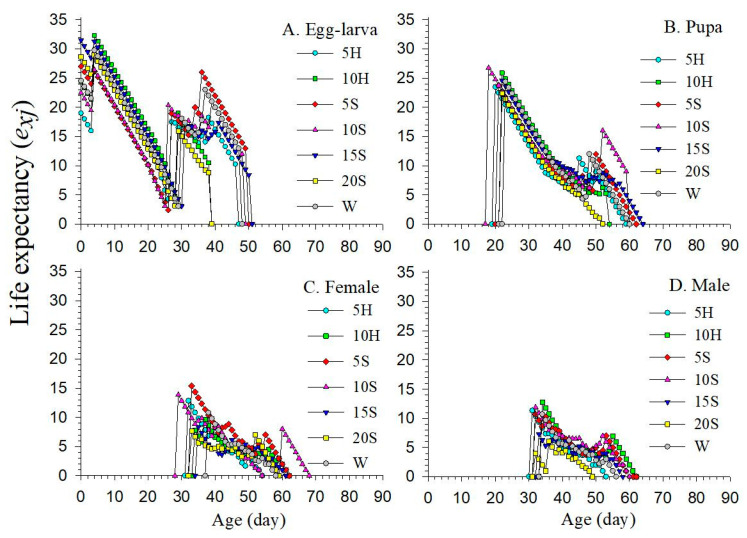
Life expectancy of F_1_ *C. sasakii* produced from F_0_ adults fed with seven different nutrient solutions. (**A**) Life expectancy of egg–larva, (**B**) life expectancy of pupa, (**C**) life expectancy of female adults, (**D**) life expectancy of male adults. W.—double-distilled water; 5H.—5 g·L^−1^ honey solution; 10H.—10 g·L^−1^ honey solution; 5S.—5 g·L^−1^ sucrose solution; 10S.—10 g·L^−1^ sucrose solution; 15S.—15 g·L^−1^ sucrose solution; 20S.—20 g·L^−1^ sucrose solution.

**Figure 6 insects-15-00283-f006:**
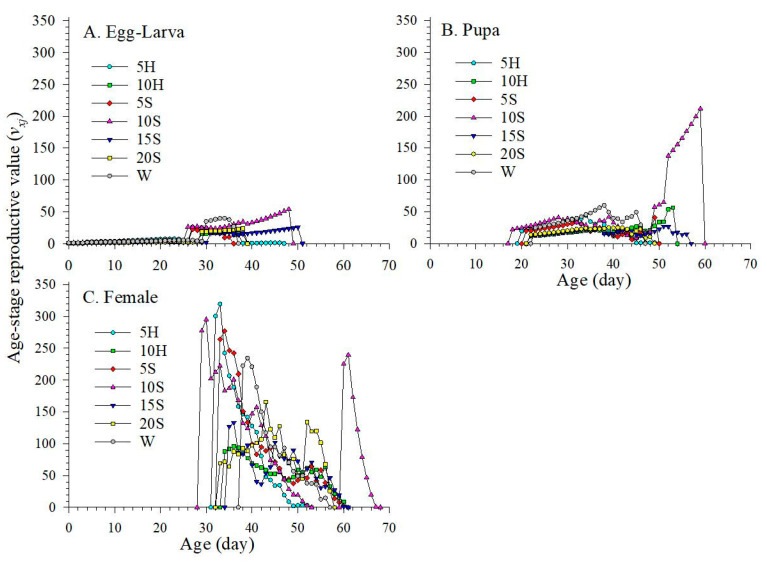
Age-stage reproductive value (*v_xj_*) of F_1_ *C. sasakii* produced from F_0_ adults fed with 7 different nutrient solutions. (**A**) Age-stage reproductive value of egg–larva, (**B**) Age-stage reproductive value of pupa, (**C**) Age-stage reproductive value of female adult. W.—double-distilled water; 5H.—5 g·L^−1^ honey solution; 10H.—10 g·L^−1^ honey solution; 5S.—5 g·L^−1^ sucrose solution; 10S.—10 g·L^−1^ sucrose solution; 15S.—15 g·L^−1^ sucrose solution; 20S.—20 g·L^−1^ sucrose solution.

**Figure 7 insects-15-00283-f007:**
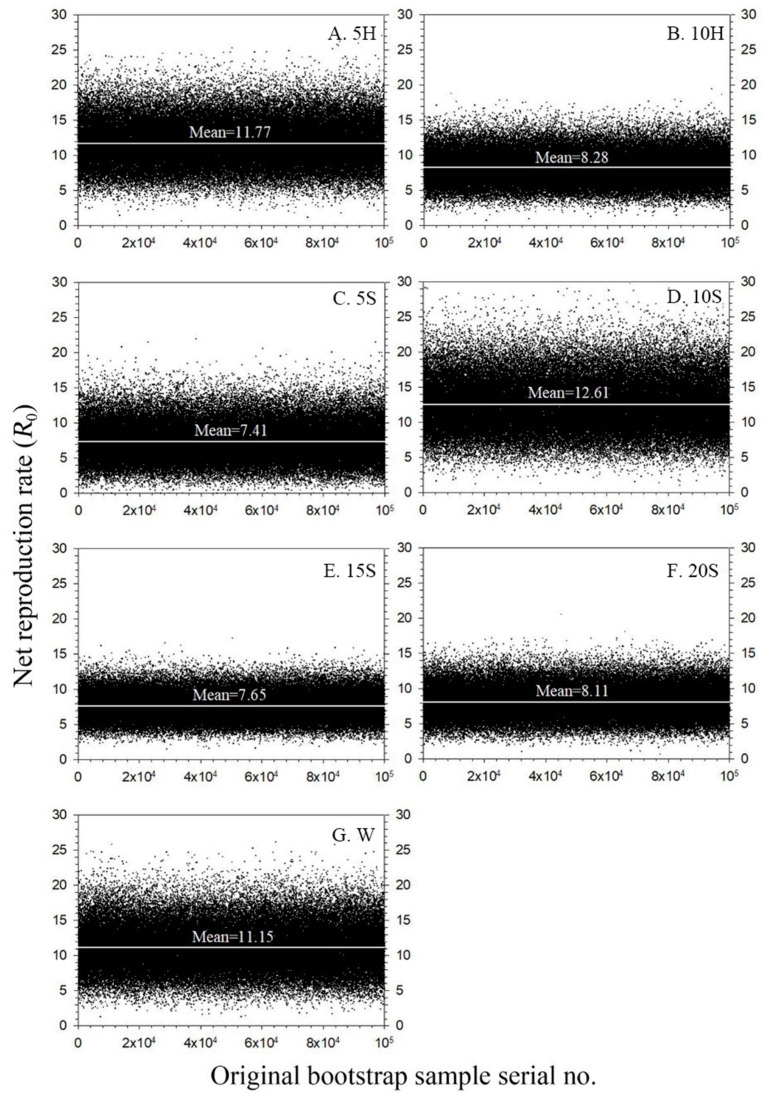
The original bootstrap *R*_0_ results of F_1_ *C. sasakii* produced from F_0_ adults fed with seven different nutrient solutions. (**A**) 5 g·L^−1^ honey solution treatment (5H), (**B**) 10 g·L^−1^ honey solution treatment (10H), (**C**) 5 g·L^−1^ sucrose solution treatment (5S), (**D**) 10 g·L^−1^ sucrose solution treatment (10S), (**E**) 15 g·L^−1^ sucrose solution treatment (15S), (**F**) 20 g·L^−1^ sucrose solution treatment (20S), (**G**) double-distilled water treatment (W).

**Figure 8 insects-15-00283-f008:**
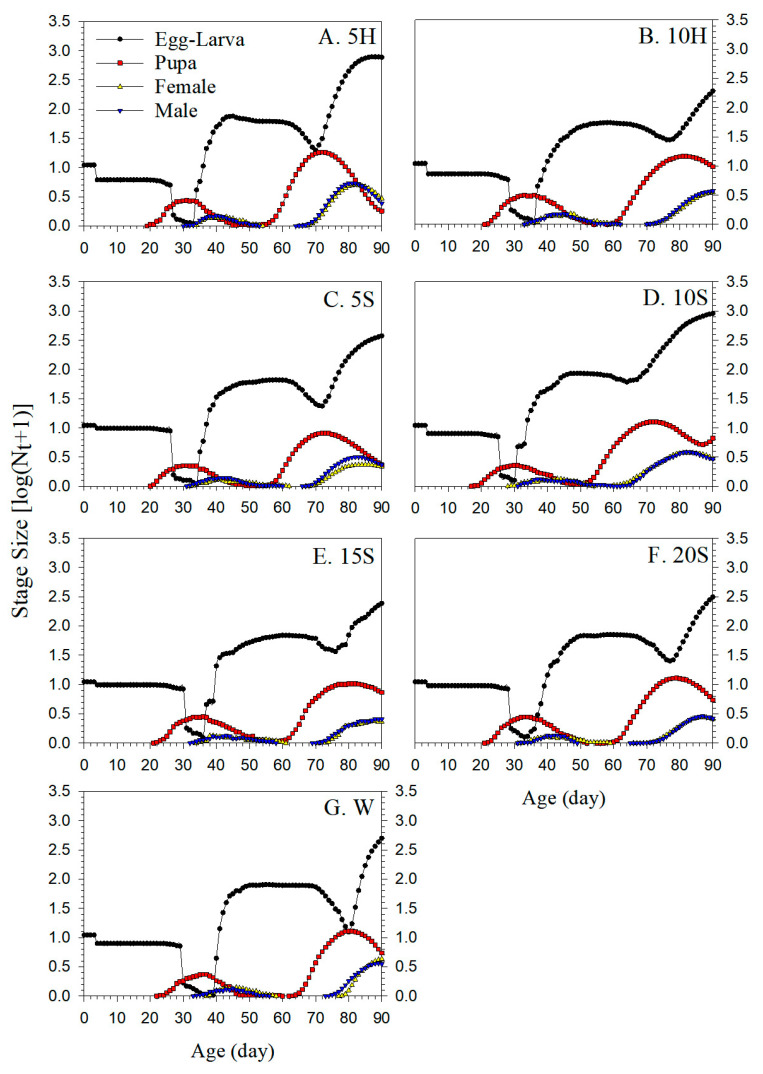
Simulation of absolute colony growth of F_1_ *C. sasakii* produced from F_0_ adults fed with seven different nutrient solutions. (**A**) 5 g·L^−1^ honey solution treatment (5H), (**B**) 10 g·L^−1^ honey solution treatment (10H), (**C**) 5 g·L^−1^ sucrose solution treatment (5S), (**D**) 10 g·L^−1^ sucrose solution treatment (10S), (**E**) 15 g·L^−1^ sucrose solution treatment (15S), (**F**) 20 g·L^−1^ sucrose solution treatment (20S), (**G**) double-distilled water treatment (W).

**Table 1 insects-15-00283-t001:** Mean (±SE) adult preoviposition period (APOP), total preoviposition period (TPOP), oviposition days (*O_d_*), and female adult fecundity (offspring) of F_1_ *C. sasakii* produced from F_0_ adults fed with seven different nutrient solutions.

Parameter	Treatment						
	W	5H	10H	5S	10S	15S	20S
F_1_ Female Adult Fecundity (eggs/female)	185.89 ± 30.33 a	176.55 ± 26.25 a	77.63 ± 16.08 b	202.18 ± 40.52 a	222.53 ± 35.80 a	88.23 ± 13.35 b	110.64 ± 18.67 b
*O_d_* (days)	5.44 ± 0.75 ab	5.89 ± 0.75 a	4.03 ± 0.57 bc	7.64 ± 1.12 a	6.71 ± 0.87 a	3.13 ± 0.26 c	4.16 ± 0.44 b
APOP (days)	1.28 ± 0.25 ab	1.00 ± 0.15 bc	1.62 ± 0.24 a	1.09 ± 0.09 b	1.18 ± 0.26 ab	1.38 ± 0.18 ab	0.63 ± 0.14 c
TPOP (days)	43.72 ± 0.86 ab	39.42 ± 1.07 c	45.10 ± 0.93 a	39.91 ± 1.47 c	39.53 ± 1.87 c	44.96 ± 1.29 a	41.05 ± 1.07 bc

Means with different letters in the same row indicate significant differences between treatments (paired bootstrap test, *p* < 0.05). Standard errors were estimated by using 100,000 bootstraps. W.—double-distilled water; 5H.—5 g·L^−1^ honey solution; 10H.—10 g·L^−1^ honey solution; 5S.—5 g·L^−1^ sucrose solution; 10S.—10 g·L^−1^ sucrose solution; 15S.—15 g·L^−1^ sucrose solution; 20S.—20 g·L^−1^ sucrose solution.

**Table 2 insects-15-00283-t002:** The population parameters of F_1_ *C. sasakii* produced from F_0_ adults fed with seven different nutrient solutions.

Parameter	Treatment						
	W	5H	10H	5S	10S	15S	20S
*r* (d^−1^)	0.0531 ± 0.0067 a	0.0606 ± 0.0069 a	0.0457 ± 0.0065 a	0.0476 ± 0.0099 a	0.0615 ± 0.0076 a	0.0449 ± 0.0058 a	0.0467 ± 0.0061 a
*λ* (d^−1^)	1.0546 ± 0.0071 a	1.0624 ± 0.0073 a	1.0467 ± 0.0068 a	1.0488 ± 0.0103 a	1.0634 ± 0.0081 a	1.0459 ± 0.0060 a	1.0478 ± 0.0064 a
*R*_0_ (offspring/individual)	11.15 ± 3.11 a	11.77 ± 3.05 a	8.28 ± 2.18 a	7.41 ± 2.62 a	12.61 ± 3.57 a	7.65 ± 1.82 a	8.11 ± 2.12 a
*T* (days)	45.40 ± 0.88 a	40.71 ± 0.74 c	46.29 ± 1.56 a	42.05 ± 1.60 abc	41.21 ± 1.61 bc	45.36 ± 1.40 ab	44.83 ± 1.03 ab

Means with different letters in the same row indicate significant differences between treatments (paired bootstrap test, *p* < 0.05). Standard errors were estimated by using 100,000 bootstraps. W.—double-distilled water; 5H.—5 g·L^−1^ honey solution; 10H.—10 g·L^−1^ honey solution; 5S.—5 g·L^−1^ sucrose solution; 10S.—10 g·L^−1^ sucrose solution; 15S.—15 g·L^−1^ sucrose solution; 20S.—20 g·L^−1^ sucrose solution.

## Data Availability

The data presented in this study are available on reasonable request from the corresponding authors.

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
