# Peer review of "Feeding Appropriate Nutrients during the Adult Stage to Promote the Growth and Development of Carposina sasakii Offspring"

_insects, 2024, doi:10.3390/insects15040283_

Round 1

Reviewer 1 Report

Comments and Suggestions for Authors

This paper has an extremely major flaw: it incorrectly says adult insects fed honey or sucrose as a "supplement," when in fact this was their only food. Those on the control received only water with no nutrients of any kind. "Supplement" means in addition to food, not as the food. This paper is not about "supplementation," but about "feeding." Perhaps the authors forgot to mention what the actual diet of the moths is. If instead the authors compared moths fed nothing but water to moths fed actual nutrients, then every use of the word supplement or supplementation must be deleting and replaced with feeding. I would have said the paper should be rejected at this point, except the results were so unexpected: feeding honey or sucrose to the parents lead to F1 adults with shorter longevity than those fed only water. That is surprising and goes against the common knowledge that starvation is bad and eating food is good. This is a major surprise: calorie restriction seems helpful to the next generation.

Another major problem: it's not clear in the introduction, methods, results, or discussion whether the "adults" the authors talk about are parents (F0) or the offspring (F1). This needs to be made extremely clear and unambiguous: use F0 and F1 everywhere you use the words "adults" unless it's obvious from context, especially if you are talking about other studies. Results about F0 adults from a paper cannot be compared to your paper's F1 results.

The overall results are mildly interesting: more sugar unsurprisingly leads to higher fitness in a sugar-feeding insect, but it's more surprising that the effects continue into the next generation. The introduction's cited references 1-3 suggest that this is all already well known information, however, reducing the novelty of this study. Perhaps worse, the introduction does not explain this any further, but wastes time explaining very obvious facts like how Lepidoptera eat sugar and how nutrition improves adult fecundity. The authors do not seem to realize what part of this study is novel and what part is extremely boring, and have focused on the wrong part.

The English is excellent: I found no spelling or grammatical errors (except in line 301, "arval"). However, the storytelling is poor. The authors do not know what is novel about this work and what is obvious. The figures are also unhelpful and need to be switched.

I wonder if the paper would benefit from being severely cut back in size into a short note instead of a full article. I make some notes with an eye to cutting below.

40-42 - I would not say this paper "lays the foundation for insect control strategy," as it is common knowledge that Lepidoptera adults drink sugar, and sugar baited traps already exist. Besides, nowhere in the paper do you actually discuss mass rearing of traps, so you cannot put it in the abstract. Delete.
52 Add the order and family of this species.
60-77 These references talk about the effect of nutritional supplementation on the adult, including fecundity. This study, however, is about the effect of adult supplementation on the next generation. That is a critical difference that needs to be made very clear and very obvious to the reader. I recommend cutting this paragraph into one sentence and one reference, no more. Then begin talking about past research on the effect of adult diet on the fitness of the offspring, without any more words about the effects on the adult. Such text is absolutely necessary in the introduction, but it is absent!
78-84 is the insect's main source of food is carbohydrate-rich food, then it's not supplementation: it's food.
103 Why is there a citation for the number of males and females used?
183-187 You need to clarify that these are the F1 adults, not the F0 parent generation adults.
Figure 1: These graphs are hard to compare. I think the data should be re-arranged: instead of showing the 4 life stages in one graph and separate by diet, I would separate by life stage and show all diets in the same graph. So have one graph showing the age-survival rate for egg-larvae, with 7 lines; and another for pupae, another for males, another for females.
Table 1: The table is useful, but not the easiest to compare the different treatments. I think a figure showing this data as 10 bar graphs would be much easier to understand and compare. So one bar graph for egg-larva duration with 7 bars [or skip that, as it would be in the improved figure 1], one bar graph for mortality, etc. Move the table to supplemental data.
Figure 2,3,etc: Again, re-arrange this to have all the experiments in one graph and the different life stages in different graphs. Also unclear is how this is different from Figure 1.
Figure 3: What is happening to the egg-larvae in 20S?
Figure 4: Wouldn't age-stage reproductive value only make sense for female adults? Also, again, I would re-arrange the data.
Figure 5: This is fine! Leave it as is, it's good.
294-299 Delete: this is introduction material, and does not belong in the discussion. Do not say the same information twice in the same paper.
308-309 The authors don't seem as surprised as I am by their own results: in previous literature, carbohydrates led to greater adult longevity (in the second generation or first?), but in your study they led to lower longevity! This is extremely interesting: use of the word "however" to highlight this would make your paper's uniqueness more obvious.
326-327 another example where it is not clear whether these are F0 or F1 adults.

Author Response

Manuscript: insects-2898746

“Appropriate nutritional supplementation during the adult stage to promote the growth and development of Carposina sasakii offspring”

Point-by-point response to comments of reviewers

Reviewer 1:

  1. This paper has an extremely major flaw: it incorrectly says adult insects fed honey or sucrose as a "supplement," when in fact this was their only food. Those on the control received only water with no nutrients of any kind. "Supplement" means in addition to food, not as the food. This paper is not about "supplementation," but about "feeding." Perhaps the authors forgot to mention what the actual diet of the moths is. If instead the authors compared moths fed nothing but water to moths fed actual nutrients, then every use of the word supplement or supplementation must be deleting and replaced with feeding. I would have said the paper should be rejected at this point, except the results were so unexpected: feeding honey or sucrose to the parents lead to F1 adults with shorter longevity than those fed only water. That is surprising and goes against the common knowledge that starvation is bad and eating food is good. This is a major surprise: calorie restriction seems helpful to the next generation.

Response. We are grateful for the detailed comments and suggestions, and will reply to the comments in order. We agreed with this comment and revised the manuscript accordingly.

  1. Another major problem: it's not clear in the introduction, methods, results, or discussion whether the "adults" the authors talk about are parents (F0) or the offspring (F1). This needs to be made extremely clear and unambiguous: use F0 and F1 everywhere you use the words "adults" unless it's obvious from context, especially if you are talking about other studies. Results about F0 adults from a paper cannot be compared to your paper's F1 results.

Response. We are grateful for the detailed comments and suggestions. We agreed with this comment and revised the manuscript accordingly.

  1. The overall results are mildly interesting: more sugar unsurprisingly leads to higher fitness in a sugar-feeding insect, but it's more surprising that the effects continue into the next generation. The introduction's cited references 1-3 suggest that this is all already well known information, however, reducing the novelty of this study. Perhaps worse, the introduction does not explain this any further, but wastes time explaining very obvious facts like how Lepidoptera eat sugar and how nutrition improves adult fecundity. The authors do not seem to realize what part of this study is novel and what part is extremely boring, and have focused on the wrong part.

Response. Thank you for the reminder, we apologize for the confusion. We agreed with this comment and the introduction section has been revised.

  1. The English is excellent: I found no spelling or grammatical errors (except in line 301, "arval"). However, the storytelling is poor. The authors do not know what is novel about this work and what is obvious. The figures are also unhelpful and need to be switched.
    Response. Thank you for your suggestion, we agreed with this comment and revised the error. We thoroughly reworked the introduction section, enhancing the novelty of our manuscript.
  2. I wonder if the paper would benefit from being severely cut back in size into a short note instead of a full article.

Response. The reviewer makes an interesting notion, and the proposed idea is good. However, ithis work would be more appropriate as a research article because the life table is large dataset including many individuals, sex composition, development time during each stage, stage differentiation, daily fecundity for each female, and hatched eggs in order to study insect population ecology (Chi et al. 2022).

Chi H, Güncan A, Kavousi A, et al. TWOSEX-MSChart: the key tool for life table research and education. Entomologia Generalis, 2022, 42(6): 845-849.

  1. I make some notes with an eye to cutting below. 40-42 - I would not say this paper "lays the foundation for insect control strategy," as it is common knowledge that Lepidoptera adults drink sugar, and sugar baited traps already exist. Besides, nowhere in the paper do you actually discuss mass rearing of traps, so you cannot put it in the abstract. Delete.

Response. We are grateful for the comment and suggestion, we agreed with this comment and deleted the wording accordingly.

  1. L52 Add the order and family of this species.

Response. We are grateful for this comment and suggestion. We agreed with this comment and revised manuscript accordingly. (Line 77)

  1. L60-77 These references talk about the effect of nutritional supplementation on the adult, including fecundity. This study, however, is about the effect of adult supplementation on the next generation. That is a critical difference that needs to be made very clear and very obvious to the reader. I recommend cutting this paragraph into one sentence and one reference, no more. Then begin talking about past research on the effect of adult diet on the fitness of the offspring, without any more words about the effects on the adult. Such text is absolutely necessary in the introduction, but it is absent!

Response. Thank you for the reminder, we apologize for the confusion. We have rewritten this part as follows (Line 51-57):

“Nutritional consumption in the adult stage plays a vital role in the development, growth, and reproduction of insects. Many species of Coleoptera, Hymenoptera, and Lepidoptera require supplementary nutrition during the adult stage to improve sexual maturity and reproduction [2]. Nutrition in the adult stage can prolong longevity and improve the fecundity of insects as well. However, previous studies have not provided conclusive evidence regarding the direct effects of adult parent (F0) nutritional status on the population parameters of first-generation offspring (F1), such as rates of increase.”

  1. L78-84 is the insect's main source of food is carbohydrate-rich food, then it's not supplementation: it's food.

Response. Thank you for your reminder, we agreed with this comment and have revised the language accordingly. (Line 68-69)

  1. L103 Why is there a citation for the number of males and females used?

Response. Thank you for the question. Insects require a suitable sex ratio for optimal reproductive capacity which should be cited. The optimal sex ratio for Carposina sasakii is 1:3 (Chang et al., 1964).

Chang, L. Y., N. X. Chang, Z. Q. Shi, and K. H. Hwang. 1964. Observation on mating and oviposition habits of adults of peach fruit moth (Carposina niponensis Wal.). Entomol. Knowl. 8: 271–273.

  1. L183-187 You need to clarify that these are the F1 adults, not the F0 parent generation adults.

Response. We are grateful for this comment and suggestion, we have revised our entire manuscript accordingly. (Line 208-211)

  1. Figure 1: These graphs are hard to compare. I think the data should be re-arranged: instead of showing the 4 life stages in one graph and separate by diet, I would separate by life stage and show all diets in the same graph. So have one graph showing the age-survival rate for egg-larvae, with 7 lines; and another for pupae, another for males, another for females.

Response. Thank you for the suggestion, we have revised the graphs accordingly.

  1. Table 1: The table is useful, but not the easiest to compare the different treatments. I think a figure showing this data as 10 bar graphs would be much easier to understand and compare. So one bar graph for egg-larva duration with 7 bars [or skip that, as it would be in the improved figure 1], one bar graph for mortality, etc. Move the table to supplemental data.

Response. Thank you for your suggestion, we have revised the data presentation accordingly.

  1. Figure 2,3,etc: Again, re-arrange this to have all the experiments in one graph and the different life stages in different graphs. Also unclear is how this is different from Figure 1.

Response. Thank you for your suggestion, we have revised the graphs accordingly.

  1. Figure 3: What is happening to the egg-larvae in 20S?

Response. Thank you for the question. The more concentrated sucrose solution may have had a negative impact on the mortality rate of egg-larvae.

  1. Figure 4: Wouldn't age-stage reproductive value only make sense for female adults? Also, again, I would re-arrange the data.

Response. Thank you for the question. According to Fisher (1930) and Chi (2014), the reproductive value (vxj) refers to the contribution of individuals in age x and stage j to the future population.

Fisher, R.A.1930.The genetical theory of natural selection: a complete variorum edition. Oxford University Press, Oxford, UK.

Tuan, S. J., Lee, C. C., & Chi, H. (2014). Population and damage projection of Spodoptera litura (F.) on peanuts (Arachis hypogaea L.) under different conditions using the age-stage, two-sex life table. Pest management science, 70(5), 805–813.

  1. Figure 5: This is fine! Leave it as is, it's good.

Response. Thank you for the affirmative feedback.

18.L294-299 Delete: this is introduction material, and does not belong in the discussion. Do not say the same information twice in the same paper.

Response. Thank you for the reminder, we apologize for the redundancy. We agreed with this comment and deleted the repeated information accordingly.

  1. L308-309 The authors don't seem as surprised as I am by their own results: in previous literature, carbohydrates led to greater adult longevity (in the second generation or first?), but in your study they led to lower longevity! This is extremely interesting: use of the word "however" to highlight this would make your paper's uniqueness more obvious.

Response. We are grateful for the detailed comment and suggestion, we agreed with this comment and revised the manuscript accordingly. (Line 382-384)

  1. L326-327 another example where it is not clear whether these are F0 or F1 adults.

Response. We are grateful for the comment and suggestion, we agreed with this comment and revised the manuscript accordingly.

Reviewer 2 Report

Comments and Suggestions for Authors

The paper deals with nutritional supplementation of an adult lepidopteran pest, showing in detail how exactly different types of feed influence offspring fitness. This is logical to expect, given that similar studies have been performed in other species of Lepidoptera. Before the paper can be published, the scientific novelty of the study has to be highlighted. The text needs to be rewritten in terms of both scientific soundness, grammar and style.

The names of the variants should be reproduced in full in the Results, so as to remind the readers what the abbreviations mean.

The term “population” is not application for a laboratory culture

L22: more suitable as compared to what?

L57: “the infestation rate to fruit” sounds odd, what does it precisely mean?

L57-58: what is implied by “infection” vs “infestation”?

L78: most adult-stage Lepidoptera – do you mean “the majority of species of Lepidoptera at adult stage”?

L79-84: introduction of this information here is not logical as the examples are already given above

L98-200: fecundity was high, indicating higher fecundity – what are you trying to say?

L201-203: why indicating different parameters (total fecundity and adult female fecundity) with the same letter (F) and why repeatedly introduce the same abbreviation?

L219: “from ages 33 to 37 d” doesn’t seem to be a proper designation of a range

L307 and further: The genus epithet has been introduced in full earlier and should be contracted

L359-366: the term “population growth” is repeated five times over this tiny paragraph

Comments on the Quality of English Language

L60: no need to use “different” in this phrase

L62-66: avoid multiple repetition of the same term, such as “nutritional supplementation”

L72 and elsewhere: “longer longevity” and “prolong the longevity” sound like a tautology

L185: longevity were – an apparent mixture of singular vs plural forms

L216: consider introduction of the definite article

L226-228: not only provides …. but also provides – consider revision

L232: peak … were – an apparent mixture of singular vs plural forms

L339: adult … were - an apparent mixture of singular vs plural forms

Author Response

Manuscript: insects-2898746

“Appropriate nutritional supplementation during the adult stage to promote the growth and development of Carposina sasakii offspring”

Point-by-point response to comments of reviewers

Reviewer 2:

  1. The paper deals with nutritional supplementation of an adult lepidopteran pest, showing in detail how exactly different types of feed influence offspring fitness. This is logical to expect, given that similar studies have been performed in other species of Lepidoptera. Before the paper can be published, the scientific novelty of the study has to be highlighted. The text needs to be rewritten in terms of both scientific soundness, grammar and style.

Response. We are grateful for the detailed comments and suggestions, and will reply to the comments in order. We agreed with this comment and revised the manuscript accordingly.

  1. The names of the variants should be reproduced in full in the Results, so as to remind the readers what the abbreviations mean.

Response. We are grateful for the detailed comments and suggestions, we agreed with this comment and revised manuscript accordingly.

  1. The term “population” is not application for a laboratory culture.

Response. Thank you for the reminder, we apologize for the confusion. We agreed with this comment and have changed "population" to “colony.”

  1. L22: more suitable as compared to what?

Response. Thank you for the reminder, we apologize for the confusion. we have revised in our manuscript as “The F1 offspring of F0 adult C. sasakii fed with 10 g·L-1 sucrose had significantly higher life table parameters than that of other treatments, thus 10 g·L-1 sucrose was more suitable for raising C. sasakii in laboratory than other treatments.” (Line 21-24)

  1. L57: “the infestation rate to fruit” sounds odd, what does it precisely mean?

Response. Thank you for your question. Fruit infestation rate refers to the proportion of fruits that are affected or damaged by pests or insects within a given sample. (Line 83-84)

  1. L57-58: what is implied by “infection” vs “infestation”?

Response. Thank you for the reminder, we apologize for the confusion. We have revised it accordingly. (Line 83-84)

  1. L78: most adult-stage Lepidoptera – do you mean “the majority of species of Lepidoptera at adult stage”?

Response. Thank you for the reminder, we apologize for the confusion, and we have revised it accordingly. (Line 68)

  1. L79-84: introduction of this information here is not logical as the examples are already given above.

Response. Thank you for the reminder, we apologize for the confusion. We have integrated and revised this part in the introduction.

  1. L198-200: fecundity was high, indicating higher fecundity – what are you trying to say?

Response. Thank you for the reminder, we apologize for the confusion. We revised manuscript accordingly. (Line 241-242)

  1. L201-203: why indicating different parameters (total fecundity and adult female fecundity) with the same letter (F) and why repeatedly introduce the same abbreviation?

Response. This is really a good question, we apologize for the confusion. We modified our manuscript accordingly. (Line 244-247)

  1. L219: “from ages 33 to 37 d” doesn’t seem to be a proper designation of a range

Response. Thank you for the reminder. We have revised manuscript accordingly.

  1. L307 and further: The genus epithet has been introduced in full earlier and should be contracted

Response. Thank you for your suggestion, we agreed with this comment and revised manuscript accordingly.

13 L359-366: the term “population growth” is repeated five times over this tiny paragraph

Response. Thank you for the reminder, we apologize for the repetition. We have revised this paragraph accordingly. (Line 441-450)

  1. L60: no need to use “different” in this phrase

Response. Thank you for your suggestion, we agree with your comment and have deleted this word.

  1. L62-66: avoid multiple repetition of the same term, such as “nutritional supplementation”

Response. Thank you for your reminder, we have revised it accordingly. (Line 51-57)

  1. L72 and elsewhere: “longer longevity” and “prolong the longevity” sound like a tautology

Response. Thank you for the reminder, we apologize for the confusion and we have revised it accordingly.

  1. L185: longevity were – an apparent mixture of singular vs plural forms

Response. Thank you for the reminder, we have checked our manuscript thoroughly and revised it. (Line 208-211)

  1. L216: consider introduction of the definite article

Response. We are grateful for the comment and suggestion, we agreed with this comment and revised the manuscript accordingly. (Line 274)

  1. L226-228: not only provides …. but also provides – consider revision

Response. Thank you for the suggestion, we agreed with this comment and revised our manuscript as follows. (Line 223-225)

“Age-stage-specific survival rate (sxj) provides not only a detailed description of the survival probability of newly laid eggs at age x and stage j, but also a comprehensive account of stage differentiation.”

  1. L232: peak … were – an apparent mixture of singular vs plural forms

Response. Thank you for your suggestion, we have checked our manuscript thoroughly and revised it. (Line 229-231)

  1. L339: adult … were - an apparent mixture of singular vs plural forms.

Response. Thank you for your suggestion, we have checked our manuscript thoroughly and revised it. (Line 422)

Reviewer 3 Report

Comments and Suggestions for Authors

A brief summary

 Carposina sasakii is an important pest in orchards in several Asian countries including China, Korea, Japan and others. The authors of this manuscript explored nutritional supplementation to improve laboratory mass-rearing of C. sasakii and studied food-baited mass trapping against C. sasakii. The manuscript was well-written. The methodology was clearly defined with good references and it addressed the objectives of the study. Statistical methods were appropriate. Results of the study were well presented in the form of tables and graphs.  The authors discussed the nutritional aspects of the various supplementation and provided some examples of what has been done to other insects.

Specific comments

There are only a few minor suggestions to improve the quality of the manuscript:

Introduction

Line 79

… supplementation, such the plant … à should this be “such as the plant …”

Materials and Methods

Lines 94-101

The authors mentioned that C. sasakii has been continuously raised in the lab for over 30 generations. Have field insects been added during that period of time? Adding field insects plays role in improving colony performance. How do you ensure the lab colony performed well in the field? Are there quality control measures to check? 

Results

Additional information needed (legends) on each Figure and Table à what are 5H, 5S, 10H etc.

Author Response

Manuscript: insects-2898746

“Appropriate nutritional supplementation during the adult stage to promote the growth and development of Carposina sasakii offspring”

Point-by-point response to comments of reviewers

Review 3

Carposina sasakii is an important pest in orchards in several Asian countries including China, Korea, Japan and others. The authors of this manuscript explored nutritional supplementation to improve laboratory mass-rearing of C. sasakii and studied food-baited mass trapping against C. sasakii. The manuscript was well-written. The methodology was clearly defined with good references and it addressed the objectives of the study. Statistical methods were appropriate. Results of the study were well presented in the form of tables and graphs.  The authors discussed the nutritional aspects of the various supplementation and provided some examples of what has been done to other insects.

Response. Thank you for the comments.

Specific comments

  1. There are only a few minor suggestions to improve the quality of the manuscript:

Introduction L 79… supplementation, such the plant … à should this be “such as the plant …”

Response. Thank you for the reminder, we have revised our manuscript thoroughly.

  1. Materials and Methods. Lines 94-101. The authors mentioned that C. sasakiihas been continuously raised in the lab for over 30 generations. Have field insects been added during that period of time? Adding field insects plays role in improving colony performance. How do you ensure the lab colony performed well in the field? Are there quality control measures to check?

Response. This is really a good question. In fact, field individuals from pesticide-free orchards were added to the rearing population an-nually to minimize inbreeding depression.

  1. Results. Additional information needed (legends) on each Figure and Table à what are 5H, 5S, 10H etc.

Response. We are grateful for the detailed comments and suggestions, we agreed with this comment and revised manuscript accordingly.

Round 2

Reviewer 1 Report

Comments and Suggestions for Authors

The corrections are a big improvement! Thank you.

Figure captions should be one paragraph, so delete the line breaks.

Comments on the Quality of English Language

Some comments
89-90 "although the adult does not intake nutrition in the field is under reported" does not make grammatical sense. Please rephrase.
101 replace "strategy, against" with "strategy for"

Author Response

Manuscript: insects-2898746

“Feeding appropriate nutrients during the adult stage to promote the growth and development of Carposina sasakii offspring”

Point-by-point response to comments of reviewer 1.

Reviewer 1

  1. The corrections are a big improvement! Thank you.

Response. Thank you for the comment.

  1. Figure captions should be one paragraph, so delete the line breaks.

Response. Thank you for the reminder, we have revised the manuscript accordingly.

  1. 89-90 "although the adult does not intake nutrition in the field is under reported" does not make grammatical sense. Please rephrase.

Response. Thank you for the suggestion, we agreed with this comment and revised manuscript accordingly.

“The caloric intake of these individuals has obvious effects on the survival of the laboratory colony. Although honey solution is often used to feed adults during laboratory rearing [21], it has been reported that adult C. sasakii do not feed in the field [22].” (Line 88-90)

  1. 101 replace "strategy, against" with "strategy for".

Response. Thank you for your suggestion, we agreed with this comment and revised the manuscript accordingly.

“This study not only provides clear evidence for the implications of F0 nutritional conditions on F1 fitness in insects, but also lays the foundation for the implementation of feeding technologies within the context of a well-conceived insect laboratory rearing strategy for C. sasakii.” (Line98-101)

Reviewer 2 Report

Comments and Suggestions for Authors

The manuscript revision is sound scientifically, but the grammar & style both leave a lot to be desired (see the respective section)

In their response to the reviewer’s comments, the authors claim they have corrected the use of term “population” but it still can be found in the context of laboratory culture (e.g. L389)

L71: the Latin title of a taxon at the genus and higher level should start with a capital letter, but not the respective adjective

L147 and further: “pupa duration” is not a logically correct term

L196: what does “direct F1 offspring” mean, is there such thing as “indirect F1 offspring”?

L350: “915.63 offspring” –the measurement unit is missing

L372: what “developmental stages” stand for here? What exactly was affected?

Comments on the Quality of English Language

L58: “while” & “but” are not complimentary to each other, one should be deleted

L62: “vertebral studies” are the studies of vertebra & vertebral column, what do you mean here?

L72: “fed with treatments” sound odd, “treatments” is not a kind of feed

L73: “fed in a water control group” sonds odd again, you designate the control group name but it’s not obvious grammatically what the insects were fed with

L74: “fed high proportions” – a pretext missing

L90: “is under reported” – confusing grammar

L450: “conditions affects” a mixture of plural and singular forms of the noun and the predicate

Author Response

Manuscript: insects-2898746

“Feeding appropriate nutrients during the adult stage to promote the growth and development of Carposina sasakii offspring”

Point-by-point response to comments of reviewer 2.

Reviewer 2

  1. The manuscript revision is sound scientifically, but the grammar & style both leave a lot to be desired (see the respective section).

Response. We are grateful for the detailed comments and suggestions, and will reply to the comments in order. We agreed with this comment and revised the manuscript accordingly.

  1. In their response to the reviewer’s comments, the authors claim they have corrected the use of term “population” but it still can be found in the context of laboratory culture (e.g. L389).

Response. Thank you for the reminder, we agreed with this comment and revised manuscript accordingly.

  1. L71: the Latin title of a taxon at the genus and higher level should start with a capital letter, but not the respective adjective.

Response. Thank you for the reminder, we agreed with this comment and revised manuscript accordingly.

“Various concentrations of sucrose and honey solution are often used to study the effects of nutrition on the development and reproduction of lepidopteran F0 individuals.” (Line 71)

  1. L147 and further: “pupa duration” is not a logically correct term.

Response. Thank you for the suggestion, we agreed with this comment and revised manuscript accordingly.

“Similarly, the duration time of larvae that did not mature/emerge was recorded as the average larva duration of each treatment, and the duration time of pupae that did not survive to adulthood was recorded as the average duration time of pupae of each treatment.” (Line 149-152)

  1. L196: what does “direct F1 offspring” mean, is there such thing as “indirect F1 offspring”?

Response. Thank you for the reminder, we apologize for the confusion. We have revised the manuscript accordingly.

“Differences in the effects of feeding different nutrients to adult stage F0 C. sasakii on their F1 offspring were observed at all stages of F1 development (Fig. 1).” (Line 200-201)

  1. L350: “915.63 offspring” –the measurement unit is missing.

Response. Thank you for the reminder, we agreed with this comment and revised manuscript accordingly.

“After 90 days, the 10S treatment colony was expected to reach a total of 915.63 offspring individuals (906.02 egg-larvae, 5.67 pupae, and 3.93 adults) which was nearly twice as great as the predicted total of 506.55 W offspring individuals (496.27 egg-larvae, 4.44 pupae, and 5.84 adults).” (Line 341-345)

  1. L372: what “developmental stages” stand for here? What exactly was affected?

Response. This is really a good question, we apologize for the confusion. We have revised this sentence as follows.

“In the results of this study, the longevity, egg-larva duration, pupa duration, adult duration and fertility of F1 C. sasakii offspring were significantly affected by both the type and concentration of nutrient solutions fed to F0 parents.” (Line 364-366)

  1. L58: “while” & “but” are not complimentary to each other, one should be deleted.

Response. Thank you for the reminder. We agreed with this comment and revised the manuscript accordingly.

“While nutritional requirements become more crucial to F1 after birth, minor variations during prenatal development can have significant impacts on the phenotype and adaptability of the F1 individual [5, 6].” (Line 58-60)

  1. L62: “vertebral studies” are the studies of vertebra & vertebral column, what do you mean here?

Response. This is really a good question, we apologize for the confusion. We have revised the manuscript as follows.

“Previous research is composed of a large number of vertebrate studies, indicating an absence of conclusive evidence regarding the direct effects of F0 nutritional status on F1 fitness in insects,” (Line 62-63)

  1. L72: “fed with treatments” sound odd, “treatments” is not a kind of feed; L73: “fed in a water control group” sonds odd again, you designate the control group name but it’s not obvious grammatically what the insects were fed with.

Response. Thank you for the reminder, we apologize for the confusion. We have revised the manuscript as follows.

Plutella xylostella adults fed with carbohydrate had significantly greater longevity than those fed in a water, and honey markedly increased fecundity.” (Line 71-73)

  1. L74: “fed high proportions” – a pretext missing.

Response. Thank you for the reminder, we apologize for the confusion. We have revised the manuscript as follows.

“On the other hand, Diglyphus isaea fed high concentrations of sugar solution had significantly reduced adult longevity than those fed low concentrations of sugar solution in the same generation [15].” (Line 73-75)

  1. L90: “is under reported” – confusing grammar.

Response. Thank you for the reminder, we apologize for the confusion. We have revised the manuscript as follows.

“Although honey solution is often used to feed adults during laboratory rearing [21], it has been reported that adult C. sasakii do not feed in the field [22].” (Line 88-89)

  1. L450: “conditions affects” a mixture of plural and singular forms of the noun and the predicate.

Response. Thank you for the reminder, we apologize for the confusion. We have revised the manuscript as follows.

“Understanding how F0 nutritional conditions affect the fitness of F1 is important for the development and implementation of pest control based on feeding technologies against C. sasakii.” (Line 441-443)
